# WHEN TEST-TIME ADAPTATION MEETS SELF-SUPERVISED MODELS

## ABSTRACT

Training on test-time data enables deep learning models to adapt to dynamic environmental changes, enhancing their practical applicability. Online adaptation from source to target domains is promising but it remains highly reliant on the performance of source pretrained models. In this paper, we investigate whether test-time adaptation (TTA) methods can continuously improve models trained via self-supervised learning (SSL) without relying on source pretraining. We introduce a self-supervised TTA protocol after observing that existing TTA approaches struggle when directly applied to self-supervised models with low accuracy on the source domain. Furthermore, we propose a collaborative learning framework that integrates SSL and TTA models, leveraging contrastive learning and knowledge distillation for stepwise representation refinement. We validate our method on diverse self-supervised models, including DINO, MoCo, and iBOT, across TTA benchmarks. Extensive experiments validate the effectiveness of our approach in SSL, showing that it achieves competitive performance even without source pretraining.

## 1 INTRODUCTION

Deep neural networks (DNNs) have achieved remarkable advancements across various fields (He et al., 2016; Dosovitskiy et al., 2021; Chen et al., 2017; Redmon et al., 2016) of computer vision and are increasingly becoming a standard tool in the industry (Wang et al., 2023; Wu et al., 2024; Kerbl et al., 2023). However, the issue of performance degradation due to domain shift (Shimodaira, 2000) between training and test datasets remains an unresolved challenge, even when distributional differences appear to be minimal (Recht et al., 2018). To address this challenge, Test-Time Training (TTT) introduces a new paradigm in domain adaptation by training at test-time to address distributional shifts between training and test data (Sun et al., 2020; Liu et al., 2021; Gandelsman et al., 2022). Building on the principles of TTT, various protocols have been developed to extend its practicality. Test-Time Adaptation (TTA) further extends this idea by adapting a pretrained model to the test domain without requiring access to source data, addressing concerns related to privacy and memory constraints (Wang et al., 2021; Zhang et al., 2022; Niu et al., 2023; Lee et al., 2024), and Continual Test-Time Adaptation (CTTA) extends TTA by assuming a continuously evolving test distribution, where the model adapts sequentially over time (Wang et al., 2022; Brahma & Rai, 2023; Liu et al., 2024b; Han et al., 2025).

Despite many achievements of TTA, discussions on the pretraining model prepared using source data and corresponding labels have been limited. For example, as shown in Figure 1a, conventional TTA required a pretraining model trained on CIFAR10 (Krizhevsky et al., 2009) to adapt to CIFAR10C (i.e., corruption set), but this model did not perform well on CIFAR100C. In other words, a separate pretraining model had to be prepared for each target domain. This limitation poses challenges in terms of practical efficiency and generality.

Along with this, our study began with a simple question: "*Is the computational cost of pretraining the source model negligible compared to the adaptation process for unlabeled target data in TTA?*" We unveil the training time required for TTA methods using a pretrained source model in Figure 1b, shedding light on the overlooked cost of source domain training and bringing it into the discussion. Optimizing the pretraining process of the source model is a practical matter, especially considering that labeled source data is often unavailable or prohibitively expensive to obtain. A simple solution

is to leverage the zero-shot performance of a self-supervised model trained through Self-Supervised Learning (SSL) on large-scale datasets (Caron et al., 2021; Chen et al., 2021; Zhou et al., 2022; Cherti et al., 2023; Oquab et al., 2024). This approach enhances generalization without requiring explicit supervision from the source domain, thereby mitigating the computational burden associated with pretraining while maintaining competitive adaptation performance in target domains. Specifically, we improve computational efficiency by designing a distance-based classifier that utilizes class prototypes obtained only through forward passes.

In this paper, we conduct an empirical investigation into the effectiveness of existing TTA approaches on self-supervised models without domain-specific knowledge and explore the feasibility of applying SSL for TTA. Figure 2a and 2b show that the primary TTA approaches, Entropy Minimization (EM) (Wang et al., 2021) and Consistency Regularization (CR) (Wang et al., 2022), are not readily applicable to SSL models. EM method minimizes predictive entropy based on the observation that lower entropy indicates higher model accuracy. While it has been demonstrated to be effective for conventional TTA, its applicability remains challenging in SSL models, where low entropy does not ensure accurate predictions. Furthermore, CR approaches that leverage pseudo-labels to maintain predictive consistency also suffer from the inaccuracy of pseudo-labels based on the low domain accuracy of SSL models.

Given that the SSL model does not seamlessly extend to TTA, we introduce a novel framework called Adapt Without Source pretraining (AWS). The proposed method consists of three key components. First, contrastive learning enhances the representation capability for both source and target domains. Second, knowledge distillation preserves the generalization ability of the initial SSL model. Third, mutual learning integrates the advantages of different predictions from the SSL and target models. Figure 2c presents the TTA performance of a source model trained with supervised learning on the source domain and a self-supervised model, DINO (Caron et al., 2021). Compared to EM and CR approaches, which fail to enhance the performance of SSL models, our method demonstrates its effectiveness in improving TTA performance for SSL models. Notably, despite the initial performance gap on the target domain, our approach surpasses the source-pretrained model, highlighting the potential for advancing TTA using SSL models.

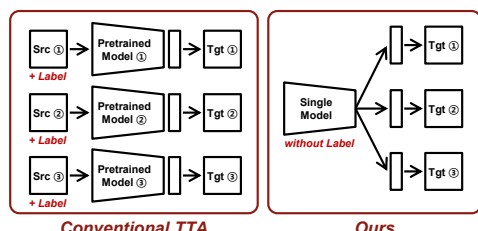

(a) Concept of Self-Supervised TTA.

| Source Model | ImageNet | CIFAR100 |
|---|---|---|
| Source Pretraining | 1h8m23s×300epochs | 9m7s×200epochs |
| SSL w/ Prototype | 36m25s | 1m25s |
| SSL w/ Prototype (Few-Shot) | 1m56s | 7s |

(b) Training time comparison.

Figure 1: **(a)** Conventional TTA methods require a separate pretraining for each source domain, whereas our Self-Supervised TTA eliminates the need for source-specific pretraining by leveraging self-supervised learning. **(b)** Training time comparison between the source pretraining of the conventional TTA and our approach.

## 2 RELATED WORK

### 2.1 TEST-TIME ADAPTATION

Distributional discrepancies between the source and target domains present a significant challenge during the deployment of DNNs (Shimodaira, 2000), and TTT introduces a learning approach that operates during test time (Sun et al., 2020). TTT mitigates domain shift by adopting supervised learning on the source domain and self-training on unlabeled target domain data (Liu et al., 2021; Gandelsman et al., 2022; Osowiechi et al., 2024). In contrast, TTA emphasizes the impracticality of accessing source domain data and instead proposes an adaptation strategy that is solely applied at test time using a source pretrained model (Wang et al., 2021). The main solution for TTA is the EM-based approach (Niu et al., 2022; 2023; Lee et al., 2024; Zhang et al., 2025a). The EM approach updates only the normalization layer and filters out inaccurate samples from the observation that samples with low entropy perform relatively well. Moreover, CTTA proposes a solution to address scenarios involving continuous domain shifts (Wang et al., 2022). CR is a primary solution in CTTA and has gained prominence for its effectiveness in stabilizing adaptation

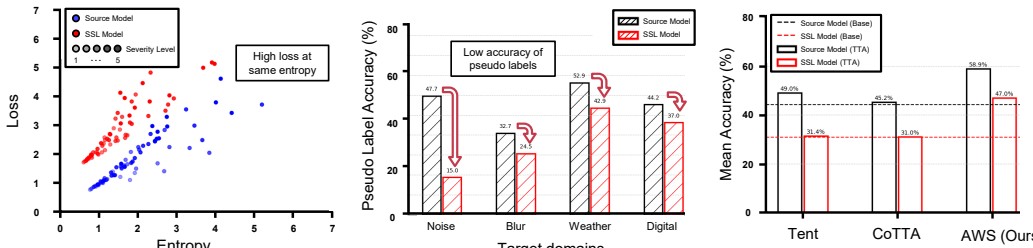

(a) Failure of the EM Method in SSL model.

(b) Failure of the CR Method in SSL model.

(c) Performance comparison in SSL model.

Figure 2: **Analysis of self-supervised models in test-time adaptation.** (a) The relationship between entropy and loss for source pretrained and SSL models. SSL models tend to exhibit higher loss for the same entropy level and may decrease the entropy of incorrect predictions, thereby increasing the true risk. (b) The accuracy of pseudo-labels for different target domains. SSL models generate pseudo-labels with lower accuracy compared to source pretrained models, which hinders performance improvement due to the propagation of inaccurate supervision signals. (c) Comparison of accuracy across different TTA approaches. Our AWS achieves improved performance for the SSL model compared with EM (Wang et al., 2021) and CR (Wang et al., 2022) methods.

Table 1: **Comparison of different adaptation protocols.** Existing protocols require training on source images and labels ($x^s$, $y^s$) during pretraining process and adapting the model to target images ($x^t$). Self-Supervised Test-Time Adaptation is based on unlabeled images ($x^u$), which is not the source domain, and does not involve training on the source data. For source domain, only a forward pass over full or few-shot is performed, without backpropagation.

| Setting | Pretrained model | | Learning procedure | |
|---|---|---|---|---|
| | Image | Label | Training loss | Test loss (data distribution) |
| Source-Free Domain Adaptation | Yes ($x^s$) | Yes ($y^s$) | $L(x^t)$ | - |
| Test-Time Training | - | - | $L(x^s, y^s) + L(x^t)$ | $L(x^t)$ (Stationary) |
| Fully Test-Time Adaptation | Yes ($x^s$) | Yes ($y^s$) | - | $L(x^t)$ (Stationary) |
| Continual Test-Time Adaptation | Yes ($x^s$) | Yes ($y^s$) | - | $L(x^t)$ (Continually changing) |
| Self-Supervised Test-Time Adaptation | Yes ($x^u$) | No | - | $L(x^t)$ (Continually changing) |

over time (Wang et al., 2022; Brahma & Rai, 2023; Liu et al., 2024b;a). The CR approach utilizes a teacher-student framework (Tarvainen & Valpola, 2017) that updates all model parameters, enabling gradual adaptation through Exponential Moving Average (EMA) update. By leveraging pseudo labels generated by an augmented teacher model, CR enforces consistency throughout the adaptation process.

## 2.2 SELF-SUPERVISED LEARNING

The training of increasingly deeper and more complex DNNs demands large amounts of data. However, the expensive cost of human annotation presents challenges for supervised learning. SSL has been proposed as an alternative, leveraging unlabeled data for various downstream tasks (Oord et al., 2018; He et al., 2020; Chen et al., 2021; 2020; Caron et al., 2021; Zhou et al., 2022; Oquab et al., 2024). CPC (Oord et al., 2018) introduces a representation learning approach based on probabilistic contrastive learning for future prediction. MoCo (He et al., 2020) employs a memory bank and a momentum encoder to facilitate contrastive learning with a large and consistent set of negative samples. SimCLR (Chen et al., 2020) leverages strong data augmentations and a contrastive loss to maximize similarity between augmented views of the same instance. DINO (Caron et al., 2021) adopts a self-distillation and teacher-student framework with a momentum encoder. iBOT (Zhou et al., 2022) proposes a mask prediction-based SSL framework through masked image modeling.

In this paper, we empirically investigate the effectiveness of TTA strategies in practical scenarios where labels are unavailable during the source pretraining phase. Furthermore, we propose Self-Supervised TTA, which leverages an SSL model as the source model and integrates it into the TTA.

# 3 SELF-SUPERVISED TEST-TIME ADAPTATION

We begin with preliminary on the Self-Supervised TTA protocol in Section 3.1. We then detail the construction of prototype classifier within this protocol and introduce our proposed method, AWS, comprising contrastive learning, knowledge distillation, and mutual learning in Section 3.2.

## 3.1 PRELIMINARY

We briefly summarize the well-known adaptation protocols for simple comparison in Table 1, including the method replacing the source pre-training process in Figure 3, and the overview of our method is also illustrated in Figure 4.

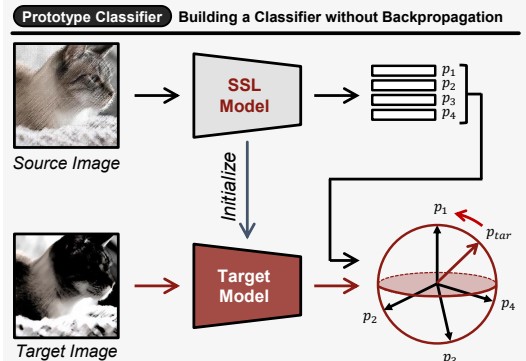

**Source Model.** Conventional TTA protocols (Wang et al., 2021; Zhang et al., 2022; Niu et al., 2023; Wang et al., 2022; Liu et al., 2024b;a) are based on supervised learning of a source model $g_s \circ f_s$ using labeled source domain data $(x^s, y^s) \in \{\mathcal{X}^s, \mathcal{Y}^s\}$, where $g_s$ and $f_s$ represent the classifier and feature extractor of the source model, respectively. Instead of requiring pretraining on the source domain, we employ a self-supervised model $f_{ssl}$ trained on an unlabeled data $x^u \in \mathcal{X}^u$. We compute feature prototypes from either a subset or the entire source dataset to align the representation of the SSL model with each class and construct a classifier $g_{ssl}$. Further details on the $g_{ssl}$ are provided in Section 3.2.

**Figure 3: A framework without source pre-training.** We construct a prototype classifier only through forward passes without a training process on the source domain ($p$ denotes prototype).

**Target Adaptation.** We follow the CTTA protocol (Wang et al., 2022), which assumes a continuously changing environment without explicit domain boundaries, to assess the adaptability of the SSL model to the target domain. The target model $g_t \circ f_t$ is initialized from the SSL model $g_{ssl} \circ f_{ssl}$. Our main objective is to adapt to the target domain by leveraging an online stream of unlabeled target data $x^t \in \mathcal{X}^t$ while minimizing the mean error as the domain gradually shifts.

## 3.2 METHODOLOGY

We briefly outline the intuition of our design. A self-supervised model offers generalizable representations but lacks source-specific knowledge; when adapted to the target domain, this limitation often leads to noisy and unreliable pseudo-labels. We aim to avoid relying solely on pseudo-labels and design a collaborative framework that leverages the SSL model's generalizable representations together with the target model's domain-specific representations.

**Prototype Classifier.** A self-supervised model typically requires a task-specific classifier to predict each class for downstream classification (Grill et al., 2020; Caron et al., 2021). Linear probing and the $k$-nearest neighbor ($k$-NN) classifier are widely used methods for building a classifier that aligns with each class (Oord et al., 2018; He et al., 2020; Chen et al., 2020). However, linear probing necessitates backpropagation for gradient computation, whereas the $k$-NN classifier entails substantial computational and memory overhead due to the requirement of storing a large number of feature representations. Inspired by the prototype-based classification in few-shot learning (Snell et al., 2017; Mensink et al., 2013) and continual learning (Rebuffi et al., 2017; Hou et al., 2019), we establish a prototype $\mu_c$ for each class $c$ and employ a cosine similarity-based classifier. Using only the forward pass enhances computational efficiency. The prediction probability for each class is given by

$$p_t(y = c|x) = \frac{\exp(\sigma \cdot \cos(f_t(x), \mu_c))}{\sum_{i \in C} \exp(\sigma \cdot \cos(f_t(x), \mu_i))}, \quad (1)$$

where $cos(\cdot, \cdot)$ denotes the cosine similarity between two vectors, $\sigma$ represents the logit scaling factor, $C$ denotes the total number of classes and $\mu_c$ is the mean of features for each class $c$ for the source dataset $\{\mathcal{X}^s, \mathcal{Y}^s\}$ of the SSL model, i.e., $\mu_c = \frac{1}{|\mathcal{X}_c^s|} \sum_{\mathcal{Y}_c^s} f_{ssl}(x^s)$.

Figure 4: **Overview of our AWS framework.** Contrastive learning refines representations by leveraging pseudo-labels while maintaining stability, knowledge distillation preserves generalization by aligning feature representations to mitigate overfitting under domain shifts, and mutual learning improves adaptation by integrating the generalization ability of the SSL model with the domain-specific knowledge of the target model through pseudo-labeling.

**Contrastive Learning.** Through a contrastive loss function, distance-based classifiers benefit from improved performance while enabling the gradual refinement of representations (Oord et al., 2018; Chen et al., 2020; Cha et al., 2021; Wen et al., 2024). Building on the need for robustness against uncertainty induced by domain shifts, we introduce an approximately correct contrastive learning method that integrates a refined segmentation of multiple prediction candidates (Zhang et al., 2024). Compensating for the low accuracy in the target domain, we identify samples sharing a pseudo label $\mathcal{T}^k$ within the top-$k$ predictions as positive samples. Conversely, when no common prediction exists among $\mathcal{T}^n$, which denotes the top-$n$ predictions with $n > k$, the sample is treated as a negative instance. For ambiguous samples that do not fit either category, contrastive loss is not applied. Accordingly, the indicator function is defined as

$$\mathbb{1}_{ij} = \begin{cases} 1, & \text{if } \mathcal{T}_i^k \cap \mathcal{T}_j^k \neq \emptyset \\ -1, & \text{if } \mathcal{T}_i^n \cap \mathcal{T}_j^n = \emptyset \ (n > k) \\ 0, & \text{otherwise.} \end{cases} \quad (2)$$

We estimate the relationships among samples predicted as positive, ambiguous, or negative using the indicator function. By applying contrastive loss to these approximately correct sample relationships, we actively leverage the initial classification capability of the SSL model while ensuring stability. The approximately correct contrastive learning loss is defined as follows:

$$\mathcal{L}_{cl} = -\sum_{i=1}^{B}\sum_{j=1}^{B} \frac{\mathbb{1}_{ij}}{\sum_{j=1}^{B}\mathbb{1}_{ij}} \log \frac{\exp(S_{ij}^t)}{\sum_{k=1}^{B}\exp(S_{ik}^t)}, \quad (3)$$

where $S_{ij}^t$ represents the cosine similarity between $f_t(x_i)$ and $f_t(x_j)$, and $B$ denotes the batch size.

**Knowledge Distillation.** As a fundamental technique for transferring knowledge between models, knowledge distillation (Hinton et al., 2015) has demonstrated effectiveness in various domains, including model compression (Romero et al., 2014; Zagoruyko & Komodakis, 2017), mitigating catastrophic forgetting (Rebuffi et al., 2017; Hou et al., 2019), improving zero-shot performance (Vemulapalli et al., 2024; Zhang et al., 2025b). To preserve generalization performance and mitigate overfitting under continuous domain shifts, we transfer knowledge from the SSL model to the target model. By reducing the discrepancy between feature representations, we retain the knowledge embedded in the SSL model while ensuring prediction consistency in the prototype classifier, which relies on cosine similarity between feature vectors and weight vectors of the classifier. To this end, we propose a knowledge distillation loss that aligns normalized feature vectors, facilitating stable knowledge transfer while preserving the geometric structure of the feature space.

$$\mathcal{L}_{kd} = \frac{1}{B}\sum_{i=1}^{B} \|\overline{f}_t(x_i) - \overline{f}_{ssl}(x_i)\|_2, \quad (4)$$

where $\overline{f}(x) = \frac{f(x)}{\|f(x)\|}$ denotes normalized feature vector, and $\| \cdot \|_2$ represents the Frobenius norm.

**Mutual Learning.** A self-supervised model demonstrates generalization performance by training on large-scale datasets, whereas a target model acquires domain-specific knowledge through adaptation. Drawing insight from studies suggesting that collaborative learning between models enhances robustness to noisy labels (Han et al., 2018; Yu et al., 2019; Wei et al., 2020), we propose a collaborative mutual learning framework to integrate the strengths of these distinct predictive tendencies. To adapt the model to the target domain, we update the SSL model's classifier using pseudo labels generated by the target model, which maintains relatively high accuracy. This enables classifier refinement while preserving the fixed embeddings of the SSL model. Furthermore, we maximize the mutual information between predicted probability distributions to capture relational information between samples, leveraging the SSL model's representational capabilities. The collaborative loss for mutual knowledge transfer is formulated as follows:

$$\mathcal{L}_{ml} = \frac{1}{B} \sum_{i=1}^{B} [\underbrace{\mathcal{H}(p_i^{ssl}, \hat{p}_i^t)}_{\text{loss for SSL}} + \underbrace{I(p_i^t, p_i^{ssl})}_{\text{loss for target}}], \tag{5}$$

where $p_i^t$ denotes the probability obtained by applying the softmax function to $g_t \circ f_t(x_i)$ and $\hat{p}_i^t = \text{argmax}(p_i^t)$. $I(p, q)$ represents the mutual information (Ji et al., 2019), and $\mathcal{H}(p, q)$ is cross entropy between two probability distributions $p$ and $q$.

The total loss function of the proposed method, which consists of approximately correct contrastive learning, knowledge distillation, and mutual learning, is formulated as follows:

$$\mathcal{L}_{aws} = \mathcal{L}_{cl} + \lambda_{kd}\mathcal{L}_{kd} + \lambda_{ml}\mathcal{L}_{ml}, \tag{6}$$

where $\lambda_{kd}$ and $\lambda_{ml}$ are hyperparameters for knowledge distillation loss and mutual loss, respectively.

## 4 EXPERIMENTS

In this section, we begin by evaluating proposed Self-Supervised TTA protocol using DINO (Caron et al., 2021), MoCo (Chen et al., 2021), and iBOT (Zhou et al., 2022). We also assess our methodology under the conventional protocol, which uses a source pretrained model. We first provide the experimental setup including the datasets, models, and the compared methods in Section 4.1. Section 4.2 describes the results for the self-supervised models and Section 4.3 for the source pretrained model.

### 4.1 EXPERIMENTAL SETUP

**Datasets and Models.** We conduct our experiments on standard CTTA benchmarks, including ImageNet-to-ImageNetC (Hendrycks & Dietterich, 2018), CIFAR10-to-CIFAR10C, and CIFAR100-to-CIFAR100C (Krizhevsky et al., 2009). ImageNetC, CIFAR10C, and CIFAR100C are corruption sets for each source data, with 15 types of 4 main categories, which serve as sequential target domains. Following (Wang et al., 2022; Liu et al., 2024b;a), we sequentially adapt the pretrained model to 15 target domains with the highest corruption level of 5 and evaluate its online prediction performance by measuring the mean error rate. Following (Liu et al., 2024b;a), we adopt ViT-B/16 (Dosovitskiy et al., 2021) as the backbone network. We present experimental results for both source pretrained and self-supervised models, using DINO (Caron et al., 2021), MoCo (Chen et al., 2021), and iBOT (Zhou et al., 2022) as SSL models.

**Compared Methods.** We compare our AWS with the well-known state-of-the-art methods: Tent (Wang et al., 2021), CoTTA (Wang et al., 2022), SAR (Niu et al., 2023), PETAL (Brahma & Rai, 2023), COME (Zhang et al., 2025a), ViDA (Liu et al., 2024b), and Continual-MAE (Liu et al., 2024a). ViDA and Continual-MAE require additional training as they incorporate an extra adapter into the source model. This makes it challenging to apply them using self-supervised models. Therefore, we do not include their results on self-supervised models.

**Implementation Details.** We employ the SGD optimizer with a momentum of 0.9 for training on the target domain. The batch size is 64 for ImageNetC and 16 for CIFAR datasets. The learning rate is set to $1e\text{-}4 \times \frac{\text{batch size}}{64}$ for the source pretrained models, and we select the range of $[1e\text{-}3, 1e\text{-}4, 1e\text{-}5, 1e\text{-}6] \times \frac{\text{batch size}}{64}$ for the self-supervised models. More implementation details in Appendix A.

Table 2: **Classification error rate (%) for ImageNet-to-ImageNetC with self-supervised models.** Mean (%) denotes the average error rate across 15 target domains. Gain (%) represents the improvement over "No Adapt". FS denotes the few-shot setup that utilizes a prototype classifier constructed with 30 samples per class. The **bold** indicates best performance.

| Pretrained Model | Method | Gaussian | shot | impulse | defocus | glass | motion | zoom | snow | frost | fog | brightness | contrast | elastic_trans | pixelate | jpeg | Mean↓ | Gain↑ |
|---|---|---|---|---|---|---|---|---|---|---|---|---|---|---|---|---|---|---|
| DINO | No Adapt | 85.7 | 83.6 | 85.7 | 68.7 | 86.5 | 73.3 | 73.4 | 64.3 | 64.3 | 61.8 | 38.1 | 79.8 | 65.7 | 55.8 | 50.8 | 69.2 | 0.0 |
| | Tent | 81.8 | 75.9 | 75.6 | 67.3 | 94.0 | 73.6 | 73.4 | 62.1 | 62.7 | 61.4 | 38.2 | 75.4 | 67.9 | 51.9 | 48.6 | 67.3 | +1.9 |
| | CoTTA | 98.2 | 99.1 | 99.3 | 68.7 | 78.7 | 72.0 | 70.9 | 69.9 | 64.9 | 61.7 | 41.0 | 78.1 | 59.8 | 52.9 | 51.8 | 71.1 | -1.9 |
| | SAR | 81.0 | 73.5 | 73.3 | 68.8 | 91.0 | 73.0 | 72.1 | 61.8 | 62.5 | 61.1 | 38.2 | 74.6 | 67.6 | 51.7 | 48.5 | 66.6 | +2.6 |
| | PETAL | 97.8 | 98.1 | 98.5 | 68.0 | 86.6 | 74.7 | 72.8 | 64.6 | 64.6 | 60.7 | 38.3 | 80.2 | 66.5 | 55.6 | 51.2 | 71.9 | -2.7 |
| | COME | 85.7 | 83.5 | 85.7 | 68.6 | 86.5 | 73.3 | 73.4 | 64.2 | 64.2 | 61.6 | 38.1 | 80.3 | 65.7 | 56.5 | 51.2 | 69.2 | +0.0 |
| | AWS | **65.9** | **59.6** | **60.7** | **57.8** | **59.3** | **57.0** | **52.7** | **50.8** | **50.9** | **50.3** | **37.0** | **52.6** | **49.6** | **45.0** | **45.6** | **53.0** | **+16.2** |
| | AWS-FS | 66.7 | 61.0 | 63.0 | 59.1 | 61.5 | 57.9 | 53.5 | 52.3 | 52.1 | 51.2 | 39.1 | 54.3 | 50.7 | 46.3 | 47.7 | 54.4 | +14.8 |
| MoCo | No Adapt | 91.2 | 89.5 | 92.1 | 79.9 | 90.2 | 79.8 | 82.6 | 74.3 | 76.4 | 80.3 | 43.1 | 85.4 | 71.2 | 52.6 | 59.6 | 76.5 | 0.0 |
| | Tent | 91.2 | 89.5 | 92.1 | 79.9 | 90.2 | 79.8 | 82.7 | 74.3 | 76.4 | 80.4 | 43.1 | 85.4 | 71.2 | 52.7 | 59.7 | 76.6 | -0.1 |
| | CoTTA | 96.9 | 94.3 | 98.1 | 80.8 | 95.6 | 82.7 | 83.8 | 74.6 | 76.1 | 78.1 | 42.9 | 86.7 | 70.9 | 52.1 | 59.0 | 78.2 | -1.7 |
| | SAR | 91.1 | 89.1 | 91.2 | 79.9 | 90.7 | 78.7 | 82.0 | 72.6 | 73.7 | 78.0 | 41.6 | 85.4 | 68.8 | 51.0 | 57.2 | 75.4 | +1.1 |
| | PETAL | 96.9 | 94.3 | 98.1 | 80.8 | 95.6 | 82.7 | 83.9 | 74.8 | 76.2 | 77.8 | 42.9 | 86.4 | 71.1 | 51.9 | 59.2 | 78.2 | -1.7 |
| | COME | 91.1 | 89.1 | 91.1 | 79.9 | 90.8 | 78.7 | 81.9 | 72.6 | 73.0 | 77.1 | **41.3** | 85.2 | 68.7 | 51.3 | 57.5 | 75.3 | +1.2 |
| | AWS | **89.4** | **81.9** | **80.1** | **71.3** | **76.5** | **70.1** | 70.5 | **61.2** | **60.7** | 63.9 | 43.8 | 62.7 | **61.4** | 48.5 | 50.2 | **66.1** | **+10.4** |
| | AWS-FS | 90.1 | 82.9 | 81.1 | 73.1 | 77.2 | 71.8 | 71.2 | 62.7 | 62.6 | 64.9 | 46.0 | 63.6 | 62.2 | 51.0 | 51.7 | 67.4 | +9.1 |
| iBOT | No Adapt | 86.1 | 84.2 | 86.9 | 69.3 | 87.6 | 74.6 | 73.3 | 62.3 | 62.5 | 60.3 | 36.1 | 78.5 | 62.2 | 48.9 | 47.2 | 68.0 | 0.0 |
| | Tent | 86.1 | 84.0 | 87.2 | 68.8 | 88.4 | 71.3 | 71.2 | 60.5 | 61.3 | 60.3 | 36.3 | 79.4 | 63.2 | 47.1 | 48.0 | 67.5 | +0.5 |
| | CoTTA | 86.1 | 84.3 | 87.0 | 69.3 | 87.6 | 77.3 | 73.3 | 61.8 | 61.9 | 60.0 | 36.1 | 78.0 | 61.9 | 48.4 | 46.7 | 68.0 | +0.0 |
| | SAR | 85.7 | 83.2 | 85.1 | 68.8 | 87.9 | 70.9 | 71.3 | 60.0 | 61.1 | 60.3 | 36.2 | 78.3 | 62.7 | 47.1 | 47.7 | 67.1 | +0.9 |
| | PETAL | 86.1 | 84.3 | 87.0 | 69.3 | 87.6 | 77.3 | 73.3 | 61.6 | 61.8 | 59.9 | 36.0 | 77.9 | 61.9 | 48.3 | 46.7 | 67.9 | +0.1 |
| | COME | 86.2 | 84.2 | 87.0 | 69.2 | 87.6 | 74.5 | 73.3 | 62.4 | 62.5 | 60.3 | 36.2 | 78.4 | 66.2 | 48.9 | 47.1 | 68.0 | +0.0 |
| | AWS | **56.4** | **51.5** | **53.4** | **53.3** | **55.0** | **52.5** | **48.5** | **46.3** | **48.1** | **46.6** | **34.8** | **47.4** | **44.6** | **40.5** | **42.8** | **48.1** | **+19.9** |
| | AWS-FS | 58.2 | 53.3 | 55.2 | 55.6 | 56.0 | 54.3 | 50.8 | 48.7 | 49.7 | 49.7 | 36.4 | 49.8 | 45.8 | 42.3 | 44.3 | 49.9 | +18.1 |

Table 3: **Summary of mean classification error (%) on CIFAR10C and CIFAR100C with self-supervised models.** The number of parentheses indicate the performance gain over "No Adapt".

| Pretrained Model | DINO | | MoCo | | iBOT | |
|---|---|---|---|---|---|---|
| Method | CIFAR10C | CIFAR100C | CIFAR10C | CIFAR100C | CIFAR10C | CIFAR100C |
| No Adapt | 44.3 (0.0) | 64.1 (0.0) | 42.2 (0.0) | 64.2 (0.0) | 48.0 (0.0) | 65.6 (+0.0) |
| Tent | 43.5 (+0.8) | 62.9 (+1.2) | 42.7 (-0.5) | 64.4 (-0.2) | 45.8 (+2.2) | 53.3 (+12.3) |
| CoTTA | 44.3 (+0.0) | 64.1 (+3.0) | 42.2 (+0.0) | 64.3 (-0.1) | 46.6 (+1.4) | 65.2 (+0.4) |
| SAR | 43.2 (+1.1) | 54.9 (+9.2) | 42.2 (+0.0) | 64.2 (+0.0) | 40.2 (+7.8) | 51.2 (+14.4) |
| PETAL | 36.4 (+7.9) | 60.2 (+3.9) | 42.2 (+0.0) | 64.6 (-0.4) | 46.0 (+2.0) | 56.3 (+9.3) |
| COME | 42.6 (+1.7) | 61.1 (+3.0) | 42.2 (+0.0) | 64.2 (+0.0) | 45.0 (+3.0) | 60.5 (+5.1) |
| AWS [Ours] | **26.8 (+17.5)** | **50.6 (+13.5)** | **40.7 (+1.5)** | **62.1 (+2.1)** | **30.1 (+17.9)** | **50.2 (+15.4)** |
| AWS-FS [Ours] | 28.2 (+16.1) | 52.5 (+11.6) | 43.9 (-1.7) | 64.3 (-0.1) | 31.6 (+16.4) | 51.9 (+13.7) |

## 4.2 RESULTS ON SELF-SUPERVISED MODELS

**ImageNet-to-ImageNetC.** The experimental results on ImageNetC using each self-supervised model (Caron et al., 2021; Chen et al., 2021; Zhou et al., 2022) are represented in Table 2. For "No Adapt", where each model is evaluated on the target without updates, the error rates are 69.2% (DINO), 76.5% (MoCo), and 68.0% (iBOT). With DINO, our method records 53.0%, improving over "No Adapt" by 16.2%. It records 66.1% with MoCo and 48.1% with iBOT. On ImageNetC, AWS achieves the lowest error rate among all compared methods, marking a substantial improvement. An additional explanation is provided in Appendix H.

**CIFAR10-to-CIFAR10C & CIFAR100-to-CIFAR100C.** In Table 3, we summarize the mean error rates on CIFAR benchmarks for SSL models. On CIFAR10C, the error rates for "No Adapt" are 44.3% (DINO), 42.2% (MoCo), and 48.0% (iBOT). AWS reduces them to 26.8%, 40.7%, 30.1%, corresponding to improvements of 17.5%, 1.5%, 17.9%. On CIFAR100C, our method shows 50.6%, 62.1%, and 50.2% with DINO, MoCo, and iBOT, respectively. These correspond to gains of 13.5%, 2.1%, and 15.4% over "No Adapt". AWS consistently reduces error across both benchmarks, underscoring its effectiveness under distributional shift. We provide the full results for all corruption types in Appendix H.

**Few-Shot Classifier Evaluation.** In Tables 2 and 3, we report the performance of AWS-FS below the row of AWS. The few-shot classifier is constructed from the source data using 30 images per class, and the ablation on the number of samples is presented in Appendix B. Although AWS-FS tends to show slightly lower gain than AWS, it still achieves consistently significant improvements with respect to existing methods. For instance, on CIFAR10C with iBOT (Table 3), AWS-FS records

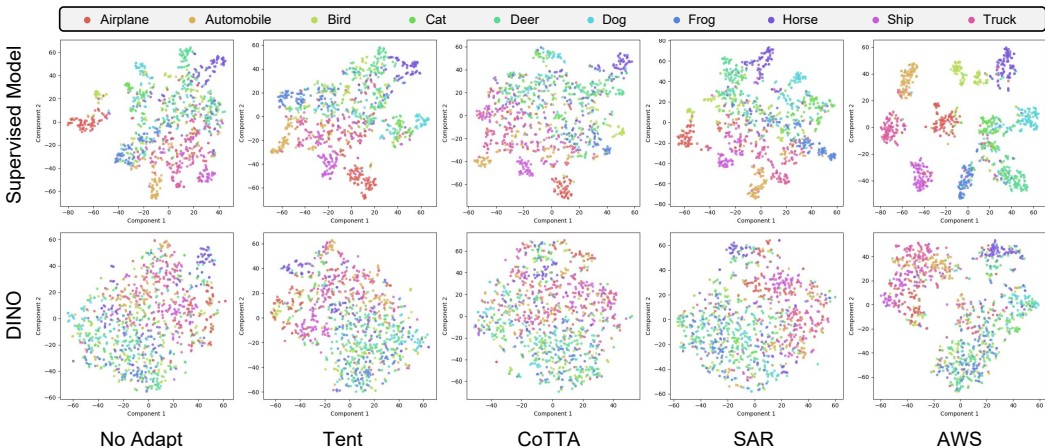

Figure 5: **Feature visualization.** We compare the t-SNE results on CIFAR10C under Gaussian noise. (above) The results of the source pretrained model. (below) The results of the SSL model, DINO.

the error rate of 31.6%, which is 1.5% higher than AWS (30.1%) yet still lower than all other baselines.

### 4.3 RESULTS ON SOURCE PRETRAINED MODEL

Table 4 presents the mean error rates on Im-ageNetC and CIFAR datasets with a source pretrained model. "No Adapt", which eval-uates the source pretrained model directly on the target, records 55.8% (ImageNetC), 28.2% (CIFAR10C), and 35.4% (CIFAR100C). On ImageNetC, we achieve the best performance of 39.4%, surpassing the prior state-of-the-art method, Continual-MAE. We achieve error rates of 10.8% on CIFAR10C and 20.4% on CI-FAR100C. Compared to the prior state-of-the-art method, we observe performance gains of 1.8% and 5.2%, respectively. Overall, AWS consistently achieves the lowest error rates on

Table 4: **Summary of mean classification error (%) with source pretrained models.** Results are reported on ImageNetC, CIFAR10C, and CI-FAR100C.

| Method | ImageNetC | CIFAR10C | CIFAR100C |
|---|---|---|---|
| No Adapt | 55.8 (0.0) | 28.2 (0.0) | 35.4 (0.0) |
| Tent | 51.0 (+4.8) | 23.5 (+4.7) | 32.1 (+3.3) |
| CoTTA | 54.8 (+1.0) | 24.6 (+3.6) | 34.8 (+0.6) |
| SAR | 45.2 (+10.6) | 26.6 (+1.6) | 26.2 (+9.2) |
| PETAL | 52.3 (+3.5) | 24.4 (+3.8) | 28.0 (+7.4) |
| ViDA | 43.4 (+12.4) | 20.7 (+7.5) | 27.3 (+8.1) |
| Continual-MAE | 42.5 (+13.3) | 12.6 (+15.6) | 26.4 (+9.0) |
| COME | 47.5 (+8.3) | 26.6 (+1.6) | 25.6 (+9.8) |
| AWS [Ours] | **39.4 (+16.4)** | **10.8 (+17.4)** | **20.4 (+15.0)** |

self-supervised models and also improves performance with source pretrained model across multi-ple benchmarks. These results demonstrate the robustness and adaptability of our method.

## 5 FURTHER ANALYSIS

**Feature Visualization.** We provide t-SNE (Van der Maaten & Hinton, 2008) visualization results to analyze the effect of TTA methods on the distribution of representations in Figure 5. After adap-tation, we extract features from the Gaussian noise corruption in CIFAR10C using both the source pretrained model and the self-supervised model, DINO. Existing approaches are typically designed to preserve the initial representations by updating only normalization layers or employing an EMA model. Consequently, these conservative update strategies demand high initial performance of the source model, leading to dependency on its initial state. In contrast, we observe that the proposed method exhibits improved decision boundaries for both the source pretrained model and the self-supervised model.

**Hyperparameter Analysis.** The proposed method involves four hyperparameters: $k$, $n$, $\lambda_{kd}$, and $\lambda_{ml}$. We conduct a grid search in Table 5 to analyze the sensitivity across all datasets using the source pretrained model. According to Table 5a, the best performing configurations of $[k, n]$ are $[1, 5]$ for ImageNetC and CIFAR10C, and $[1, 2]$ for CIFAR100C. Moreover, $\lambda_{kd}$ and $\lambda_{ml}$ represents that the best performance is obtained with $\lambda_{kd} = 0.01$ and $\lambda_{ml} = 0.4$. We observe that our method

Table 5: **AWS ablation experiments.** We investigate the sensitivity of hyperparameters in the proposed method. IN-C, C10-C, and C100-C are ImageNetC, CIFAR10C, and CIFAR100C, respectively.

(a) **Hyperparameter** $[k, n]$.

| $[k,n]$ | IN-C | C10-C | C100-C |
|---|---|---|---|
| [1, 2] | 39.5 | 11.5 | **20.4** |
| [1, 3] | 39.5 | 11.0 | 20.5 |
| [1, 5] | **39.4** | **10.8** | 20.7 |
| [3, 10] | 40.1 | 57.2 | 23.1 |
| [5, 20] | 40.4 | N/A | 24.8 |

(b) **Hyperparameter** $\lambda_{kd}$.

| $\lambda_{kd}$ | IN-C | C10-C | C100-C |
|---|---|---|---|
| 0 | 40.6 | 11.1 | 22.3 |
| 0.01 | **39.4** | **10.8** | **20.4** |
| 0.02 | 40.1 | 11.2 | 22.5 |
| 0.03 | 41.9 | 11.7 | 24.9 |
| 0.04 | 43.6 | 11.5 | 26.9 |

(c) **Hyperparameter** $\lambda_{ml}$.

| $\lambda_{ml}$ | IN-C | C10-C | C100-C |
|---|---|---|---|
| 0 | 43.8 | 13.7 | 25.1 |
| 0.1 | 41.4 | 12.8 | 23.9 |
| 0.2 | 40.3 | 11.7 | 21.5 |
| 0.3 | 39.8 | 11.6 | 20.8 |
| 0.4 | **39.4** | **10.8** | **20.4** |

Table 6: **Domain generalization** performance on ImageNetC. Results (%) are error rates on unseen domains.

| Method | Directly test on unseen domains | | | | | Unseen |
|---|---|---|---|---|---|---|
| | bri. | contrast | elastic | pixelate | jpeg | Mean↓ |
| No Adapt | 26.4 | 91.4 | 57.5 | 38.0 | 36.2 | 49.9 |
| Tent | 25.8 | 91.9 | 57.0 | 37.2 | 35.7 | 49.5 |
| CoTTA | 25.3 | 88.1 | 55.7 | 36.4 | 34.6 | 48.0 |
| ViDA | **24.6** | 68.2 | 49.8 | 34.7 | 34.1 | 42.3 |
| AWS | 24.8 | **65.9** | **47.1** | **34.1** | **33.5** | **41.1** |

Table 7: **Effect of each component**, such as Contrastive Learning (CL), Knowledge Distillation (KD), and Mutual Learning (ML).

| CL | KD | ML | IN-C | C10-C | C100-C |
|---|---|---|---|---|---|
| No Adapt [Baseline] | | | 55.8 | 28.2 | 35.4 |
| ✓ | | | 43.4 | 16.0 | 27.9 |
| | | ✓ | 42.7 | 21.3 | 21.8 |
| | ✓ | ✓ | 41.6 | 21.3 | 23.0 |
| ✓ | ✓ | | 43.8 | 14.1 | 25.1 |
| ✓ | | ✓ | 40.6 | 11.2 | 22.2 |
| ✓ | ✓ | ✓ | **39.4** | **10.8** | **20.4** |

not only exhibits low sensitivity to hyperparameters but also surpasses previous methods across a wide range of hyperparameter settings.

**Domain Generalization.** In Table 6, we evaluate the domain generalization performance on ImageNetC. Following ViDA (Liu et al., 2024b), we adapt to 10 corruption types from ImageNetC under the CTTA protocol, and subsequently evaluate performance on the 5 remaining unseen corruption types. We achieve an 8.8% improvement over No Adapt and surpasses the previous state-of-the-art by 1.2%. These results indicate that the proposed method acquires generalized knowledge and enhances representational capacity during adaptation, thereby improving performance on unseen target domains.

**Effectiveness of Individual Components.** Table 7 presents an ablation study evaluating the contribution of each component in our method, including CL, KD, and ML. First, we apply CL to enhance the representational capability of the SSL model and reduce error from 55.8% to 43.4% on ImageNetC. These results indicate that applying CL individually can effectively improve adaptation. Second, when KD is introduced to CL, we observe that it mitigates forgetting during adaptation and results in comparable or even lower mean error than using CL alone (row4). Third, adding ML to the combination of CL and KD achieves the best performance across all datasets, demonstrating that ML provides additional benefits for further performance improvement (row6). The ablation study suggests that each component contributes to complementary aspects of the adaptation process.

## 6 CONCLUSION

In this paper, we investigate the feasibility of integrating self-supervised models into TTA and explore effective strategies to enhance their adaptability within this scenario. We address the primary challenge of applying self-supervised models to TTA, the absence of a classifier, by proposing a prototype classifier without extra training and cost. Furthermore, we propose AWS, composed of CL, KD, and ML, to effectively leverage the expressive representations of self-supervised models while reducing reliance on source-specific knowledge for more stable adaptation. Extensive experiments demonstrate that our proposed AWS is highly effective not only in the self-supervised setting but also in the conventional supervised setting. Based on these results, we expect this study to contribute to expanding the potential of self-supervised models in TTA and hope that future research will build on these findings.

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

APPENDIX

The appendix provides supplementary analyses and experimental details. In Appendix A, we report the hyperparameter settings used in our experiments. We then investigate how the number of samples used to construct the few-shot classifier influences performance in Appendix B. We further extend our evaluation to natural domain shift scenarios in Appendix C, and describe the parameter update strategy of our framework in Appendix D. Appendix E includes prediction shifts after adaptation and Appendix F provides a comparison of adaptation time. We provide the semantic segmentation results in Appendix G. We offer the full results across all benchmarks with additional explanations in Appendix H. Finally, a brief note on the use of large language models (LLMs) is given in Appendix I.

## A    HYPERPARAMETERS IN EXPERIMENTS

In this section, we describe the learning rate configuration used in our experiments on SSL models. For fair comparison, we select the learning rate that yields the lowest mean error within the range defined in [1$e$-3, 1$e$-4, 1$e$-5, 1$e$-6]$\times \frac{\text{batch size}}{64}$.

1. **DINO**
   - Learning rate (ImageNetC) [**1e-3** (Tent, SAR, CoTTA), **1e-4** (PETAL, AWS), 1e-5, **1e-6** (COME)] $\times \frac{\text{batch size}}{64}$
   - Learning rate (CIFAR10C) [**1e-3** (CoTTA, PETAL), **1e-4** (Tent, SAR, COME), **1e-5** (AWS), 1e-6] $\times \frac{\text{batch size}}{64}$
   - Learning rate (CIFAR100C) [**1e-3** (SAR, PETAL), **1e-4** (Tent, COME, CoTTA), **1e-5** (AWS), 1e-6] $\times \frac{\text{batch size}}{64}$

2. **MoCo**
   - Learning rate (ImageNetC) [**1e-3** (CoTTA, PETAL), **1e-4** (SAR, COME, AWS), 1e-5, **1e-6** (Tent)] $\times \frac{\text{batch size}}{64}$
   - Learning rate (CIFAR10C) [**1e-3** (SAR, COME, CoTTA), 1e-4, **1e-5**(AWS), **1e-6** (Tent, PETAL)] $\times \frac{\text{batch size}}{64}$
   - Learning rate (CIFAR100C) [**1e-3** (CoTTA, COME, SAR), 1e-4, **1e-5**(AWS), **1e-6** (Tent, PETAL)] $\times \frac{\text{batch size}}{64}$

3. **iBOT**
   - Learning rate (ImageNetC) [1e-3, **1e-4** (Tent, SAR, CoTTA, PETAL, AWS), 1e-5, **1e-6** (COME)] $\times \frac{\text{batch size}}{64}$
   - Learning rate (CIFAR10C) [**1e-3** (SAR, CoTTA), **1e-4** (Tent, COME), 1e-5, **1e-6** (PETAL, AWS)] $\times \frac{\text{batch size}}{64}$
   - Learning rate (CIFAR100C) [**1e-3** (Tent, SAR, CoTTA, PETAL), **1e-4** (COME), 1e-5, **1e-6** (AWS)] $\times \frac{\text{batch size}}{64}$

## B    EFFECT OF THE NUMBER OF FEW-SHOT SAMPLES

We study the applicability of AWS when the classifier is constructed using a subset of the source data. To this end, we evaluate the effect of varying the number of samples ($N$) and compare it with the full-shot setting, where the entire dataset is used. Specifically, we compare the mean error rates on ImageNetC using DINO backbone across three different seeds. As shown in Figure 6, the performance is close to that achieved using the entire dataset as $N$ increases. Notably, $N = 30$ few-shot setting requires only 3% of the source data compared to the full-shot setting. Nevertheless, the performance gap to the full-shot setting remains within 1%. We observe that a classifier constructed with a subset of data can achieve performance close to the full-shot classifier. This observation suggests that our method has the potential to maintain its effectiveness even in more challenging scenarios.

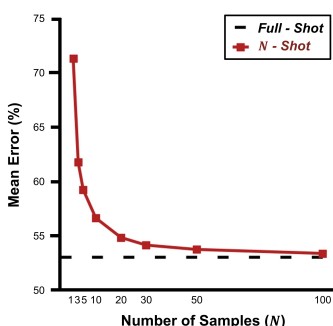

Figure 6: Effect of # of samples.

## C  NATURAL DOMAIN SHIFT SCENARIOS

In Table 8, we provide experiments on natural domain shift datasets to extend the validity of the proposed method in real-world natural changes. We conduct on ImageNet-R (Hendrycks et al., 2021), V2 (Recht et al., 2019), Sketch (Wang et al., 2019) as a target for ImageNet (Deng et al., 2009) source. We compare the results of our method with Tent (Wang et al., 2021), CoTTA (Wang et al., 2022), SAR (Niu et al., 2023) and FOA (Niu et al., 2024). Our method consistently achieves improved performance, demonstrating the effectiveness of AWS.

Table 8: Classification accuracy (%) for ImageNet-to-ImageNet-R/V2/Sketch.

| Method | Source Pretrained | | | DINO | | | MoCo | | | iBOT | | |
|---|---|---|---|---|---|---|---|---|---|---|---|---|
| | R | V2 | Sketch | R | V2 | Sketch | R | V2 | Sketch | R | V2 | Sketch |
| No Adapt | 59.5 | **75.4** | 44.9 | 39.3 | 56.9 | 24.3 | 26.6 | 52.8 | 18.3 | 40.6 | 59.1 | 25.0 |
| TENT | 63.9 | 75.2 | 49.1 | 39.6 | 57.0 | 24.4 | 26.6 | 53.2 | 18.3 | 41.3 | 59.1 | 25.1 |
| CoTTA | 63.5 | **75.4** | 50.0 | 39.5 | 57.0 | 25.5 | 21.7 | 52.1 | 11.2 | 41.1 | 59.2 | 26.8 |
| SAR | 63.3 | 75.1 | 48.7 | 39.8 | 57.0 | 24.5 | **38.0** | **54.2** | 19.0 | 41.4 | 59.0 | 25.1 |
| FOA | 63.8 | **75.4** | **54.4** | 42.1 | 57.1 | 31.5 | 20.6 | 52.6 | 8.1 | 44.4 | 59.0 | 35.6 |
| AWS [Ours] | **69.3** | **75.4** | **54.4** | **45.0** | **57.5** | **38.0** | 35.8 | 53.5 | **25.6** | **48.8** | **59.6** | **40.0** |

## D  PARAMETER UPDATE STRATEGY

We use both SSL model and target model in our framework. The SSL model, trained on large-scale datasets, ensures generalization performance, whereas the target model initialized from it acquires domain-specific knowledge through adaptation. While maintaining the generalized feature representations of the SSL model, we intend to improve the classifier through pseudo labels of the target model that has relatively high accuracy. In AWS, the encoder $f_{ssl}$ is kept frozen during adaptation and the classifier $g_{ssl}$ is updated. Table 9 shows EMA updates for $f_{ssl}$ and a fixed classifier $g_{ssl}$.

Table 9: Effect of parameter update strategy for SSL models.

| Method | Source Pretrained↓ | DINO↓ | MoCo↓ | iBOT↓ |
|---|---|---|---|---|
| $g_{ssl}$ (Frozen) | 39.9 | **53.0** | 73.0 | **48.1** |
| $f_{ssl}$ (Update) | 40.0 | 54.5 | 70.5 | 49.7 |
| AWS [Ours] | **39.4** | **53.0** | **69.5** | **48.1** |

## E  PREDICTION SHIFTS AFTER ADAPTATION

Figure 7 illustrates the change in predictions during adaptation, based on the initial predictions of the source model. The results demonstrate that our method significantly improves predictions on samples initially misclassified by the source model. We interpret this as evidence that our representation learning-based approach is less affected by confirmation bias and more effective at improving the initial model.

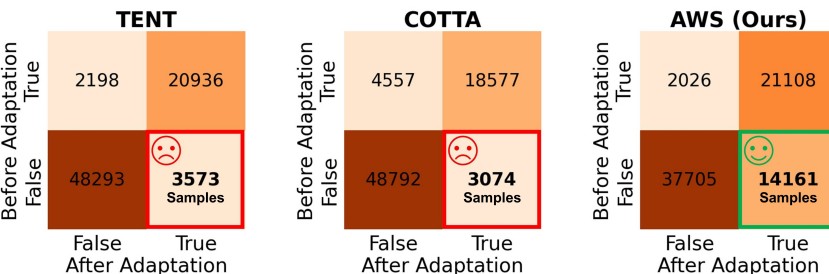

Figure 7: Prediction shifts after adaptation with respect to the source model's initial predictions.

## F ADAPTATION TIME COMPARISON

We compare the adaptation time for 15 domains from ImageNet-to-ImageNetC in Table 10. Our concern about presenting the comparison of adaptation times in the paper is that the adaptation scenarios may depend on the experimental setup, and we emphasize the efficiency of the pretraining time.

Table 10: Comparison of adaptation cost.

| Protocol | Source Pretraining Time | Target Adaptation Time (15 domains) | Total Time↓ |
|---|---|---|---|
| TTA | 1h8m23s×300epochs | 8m31s (Tent) – 54m17s (ViDA) | ≤ 342h9m17s |
| SSTTA | 1m56s (Few-shot) – 36m15s (Full) | 15m30s (Ours) | ≤ 51m45s |

## G EFFECTIVENESS ON SEMANTIC SEGMENTATION

In Table 11, we present semantic segmentation CTTA results on the Cityscapes-to-ACDC benchmark (Cordts et al., 2016; Sakaridis et al., 2021). The four weather conditions (fog, night, rain, and snow) serve as targets over three sequential rounds and performance is summarized by the overall mean mIoU. We compare our method with Tent (Wang et al., 2021), CoTTA (Wang et al., 2022), SVDP (Yang et al., 2024), and Hybrid-TTA (Park et al., 2025). While our method is devised for classification, we report the results without adding any segmentation-specific losses or modules, leaving room for further improvements via task-tailored components.

Table 11: Comparison of performance on Cityscapes-to-ACDC under CTTA scenario.

| Time | $t$ | | | | | | | | | | | | | |
|---|---|---|---|---|---|---|---|---|---|---|---|---|---|---|
| Round | 1 | | | | 2 | | | | 3 | | | | Mean↑ | Gain |
| Method | Fog | Night | Rain | Snow | Fog | Night | Rain | Snow | Fog | Night | Rain | Snow | | |
| Source | 69.1 | 40.3 | 59.7 | 57.8 | 69.1 | 40.3 | 59.7 | 57.8 | 69.1 | 40.3 | 59.7 | 57.8 | 56.7 | / |
| Tent | 69.0 | 40.2 | 60.1 | 57.3 | 68.3 | 39.0 | 60.1 | 56.3 | 67.5 | 37.8 | 59.6 | 55.0 | 55.7 | -1.0 |
| CoTTA | 70.9 | 41.2 | 62.4 | 59.7 | 70.9 | 41.1 | 62.6 | 59.7 | 70.9 | 41.0 | 62.7 | 59.7 | 58.6 | +1.9 |
| CoTTA + Ours | 71.4 | 43.3 | 65.2 | 63.1 | 73.1 | 44.3 | 66.9 | 63.2 | 72.7 | 44.3 | 67.2 | 63.3 | 61.5 | +4.8 |
| SVDP | 72.1 | 44.0 | 65.2 | 63.0 | 72.2 | 44.5 | 65.9 | 63.5 | 72.1 | 44.2 | 65.6 | 63.6 | 61.1 | +4.4 |
| SVDP + Ours | 71.8 | 44.7 | 66.6 | 63.0 | 72.5 | 45.6 | 68.0 | 63.7 | 71.3 | 43.2 | 66.8 | 63 | 61.7 | +5.0 |
| Hybrid-TTA | 70.3 | 44.5 | 65.1 | 63.2 | 71.8 | 48.2 | 67.1 | 63.7 | 71.2 | 49.3 | 67.1 | 63.3 | 62.1 | +5.4 |
| Hybrid-TTA + Ours | 70.2 | 44.2 | 65.0 | 63.0 | 72.4 | 47.5 | 66.5 | 64.1 | 72.2 | 47.3 | 66.4 | 64.4 | 62.3 | +5.6 |

## H FULL-RESULTS ON IMAGENETC, CIFAR10C, AND CIFAR100C

We provide full-results for ImageNet-to-ImageNetC, CIFAR10-to-CIFAR10, and CIFAR100-to-CIFAR100C in Tables 12, 13 and 14, respectively. Each table includes 15 corruption types for each source dataset. The error is measured in an online manner under sequential target domains. The number in parentheses under "Pretrained Model" indicate the accuracy on source domain. For ImageNetC, the source pretrained model attains 83.6% source accuracy, but the self-supervised models fall notably, recording 63.1% (DINO), 60.6% (MoCo), and 65.9% (iBOT). The lack of source specific knowledge in a SSL model results in higher initial target error (No Adapt)—69.2% (DINO), 76.5% (MoCo), and 68.0% (iBOT)—compared with 55.8% for the source pretrained model. With limited source specific knowledge and low initial performance, existing TTA methods struggle to achieve significant gains on SSL models. For example, Tent improves over "No Adapt" by 4.8% on the source pretrained model but only 1.9%, -0.1%, and 0.5% on DINO, MoCo, and iBOT, respectively. Meanwhile, AWS achieves significant improvements over "No Adapt", with 16.4% (source pretrained), 16.2% (DINO), 10.4% (MoCo), and 19.9% (iBOT), showing consistent gains across both source pretrained and self-supervised models.

## I POLICY ON LARGE LANGUAGE MODELS

We only used ChatGPT for minor English editing and language polishing of the manuscript.

Table 12: Full-results for ImageNet-to-ImageNetC under CTTA scenario. Mean (%) denotes the average error rate across 15 target domains.

| Pretrained Model (Src Acc. %) | Method | Gaussian | shot | impulse | defocus | glass | motion | zoom | snow | frost | fog | brightness | contrast | elastic_trans | pixelate | jpeg | Mean↓ | Gain↑ |
|---|---|---|---|---|---|---|---|---|---|---|---|---|---|---|---|---|---|---|
| Source pretrained (Acc. 83.6%) | No Adapt [Baseline] | 53.0 | 51.8 | 52.1 | 68.5 | 78.8 | 58.5 | 63.3 | 49.9 | 54.2 | 57.7 | 26.4 | 91.4 | 57.5 | 38.0 | 36.2 | 55.8 | 0.0 |
| | Tent [ICLR'21] | 52.2 | 48.9 | 49.2 | 65.8 | 73.0 | 54.5 | 58.4 | 44.0 | 47.7 | 50.3 | 23.9 | 72.8 | 55.7 | 34.4 | 33.9 | 51.0 | +4.8 |
| | CoTTA [CVPR'22] | 52.9 | 51.6 | 51.4 | 68.3 | 78.1 | 57.1 | 62.0 | 48.2 | 52.7 | 55.3 | 25.9 | 90.0 | 56.4 | 36.4 | 35.2 | 54.8 | +1.0 |
| | SAR [ICLR'23] | 49.3 | 43.8 | 44.9 | 58.2 | 60.9 | 46.1 | 51.8 | 41.3 | 44.1 | 41.8 | 23.8 | 57.2 | 49.9 | 32.9 | 32.7 | 45.2 | +10.6 |
| | PETAL [CVPR'23] | 52.1 | 48.2 | 47.5 | 66.8 | 74.0 | 56.7 | 59.7 | 46.8 | 47.2 | 52.7 | 26.4 | 91.3 | 50.7 | 32.3 | 32.0 | 52.3 | +3.5 |
| | ViDA [ICLR'24] | 47.7 | 42.5 | 42.9 | 52.2 | 56.9 | 45.5 | 48.9 | 38.9 | 42.7 | 40.7 | 24.3 | 52.8 | 49.1 | 33.5 | 33.1 | 43.4 | +12.4 |
| | Continual-MAE [CVPR'24] | 46.3 | 41.9 | 42.5 | 51.4 | 54.9 | 43.3 | 40.7 | 34.2 | 35.8 | 64.3 | 23.4 | 60.3 | 37.5 | 29.2 | 31.4 | 42.5 | +13.3 |
| | COME [ICLR'25] | 49.3 | 43.5 | 44.5 | 59.6 | 60.1 | 49.4 | 52.4 | 41.6 | 43.6 | 44.3 | 24.1 | 89.1 | 45.9 | 32.4 | 32.5 | 47.5 | +8.3 |
| | AWS [Ours] | 43.9 | 39.6 | 41.3 | 48.9 | 47.7 | 42.2 | 42.9 | 35.8 | 37.3 | 39.7 | 23.6 | 49.8 | 37.5 | 30.9 | 30.3 | 39.4 | +16.4 |
| DINO (Acc. 63.1%) | No Adapt [Baseline] | 85.7 | 83.6 | 85.7 | 68.7 | 86.5 | 73.3 | 73.4 | 64.3 | 64.3 | 61.8 | 38.1 | 79.8 | 65.7 | 55.8 | 50.8 | 69.2 | 0.0 |
| | Tent [ICLR'21] | 81.8 | 75.9 | 75.6 | 67.3 | 94.0 | 73.6 | 73.4 | 62.1 | 62.7 | 61.4 | 38.2 | 75.4 | 67.9 | 51.9 | 48.6 | 67.3 | +1.9 |
| | CoTTA [CVPR'22] | 98.2 | 99.1 | 99.3 | 68.7 | 78.7 | 72.0 | 70.9 | 69.9 | 64.9 | 61.7 | 41.0 | 78.1 | 59.8 | 52.9 | 51.8 | 71.1 | -1.9 |
| | SAR [ICLR'23] | 81.0 | 73.5 | 73.3 | 68.8 | 91.0 | 73.0 | 72.1 | 61.8 | 62.5 | 61.1 | 38.2 | 74.6 | 67.6 | 51.7 | 48.5 | 66.6 | +2.6 |
| | PETAL [CVPR'23] | 97.8 | 98.1 | 98.5 | 68.0 | 86.6 | 74.7 | 72.8 | 64.6 | 64.6 | 60.7 | 38.3 | 80.2 | 66.5 | 55.6 | 51.2 | 71.9 | -2.7 |
| | COME [ICLR'25] | 85.7 | 83.5 | 85.7 | 68.6 | 86.5 | 73.3 | 73.4 | 64.2 | 64.5 | 61.6 | 38.1 | 80.3 | 65.7 | 56.5 | 51.2 | 69.2 | +0.0 |
| | AWS [Ours] | 65.9 | 59.6 | 60.7 | 57.8 | 59.3 | 57.0 | 52.7 | 50.8 | 50.9 | 50.3 | 37.0 | 52.6 | 49.6 | 45.0 | 45.6 | 53.0 | +16.2 |
| | AWS-FS [Ours] | 66.7 | 61.0 | 63.0 | 59.1 | 61.5 | 57.9 | 53.5 | 52.3 | 52.1 | 51.2 | 39.1 | 54.3 | 50.7 | 46.3 | 47.7 | 54.4 | +14.8 |
| MoCo (Acc. 60.0%) | No Adapt [Baseline] | 91.2 | 89.5 | 92.1 | 79.9 | 90.2 | 79.8 | 82.6 | 74.3 | 76.4 | 80.3 | 43.1 | 85.4 | 71.2 | 52.6 | 59.6 | 76.5 | 0.0 |
| | Tent [ICLR'21] | 91.2 | 89.5 | 92.1 | 79.9 | 90.2 | 79.8 | 82.7 | 74.3 | 76.4 | 80.4 | 43.1 | 85.4 | 71.2 | 52.7 | 59.7 | 76.6 | -0.1 |
| | CoTTA [CVPR'22] | 96.9 | 94.3 | 98.1 | 80.8 | 95.6 | 82.7 | 83.8 | 74.6 | 76.1 | 78.1 | 42.9 | 86.7 | 70.9 | 52.1 | 59.0 | 78.2 | -1.7 |
| | SAR [ICLR'23] | 91.1 | 89.1 | 91.2 | 79.9 | 90.7 | 78.7 | 82.0 | 72.6 | 73.7 | 78.0 | 41.6 | 85.4 | 68.8 | 51.0 | 57.2 | 75.4 | +1.1 |
| | PETAL [CVPR'23] | 96.9 | 94.3 | 98.1 | 80.8 | 95.6 | 82.7 | 83.9 | 74.8 | 76.2 | 77.8 | 42.9 | 86.4 | 71.1 | 51.9 | 59.2 | 78.2 | -1.7 |
| | COME [ICLR'25] | 91.1 | 89.1 | 91.1 | 79.9 | 90.8 | 78.7 | 81.9 | 72.9 | 73.0 | 77.1 | 41.3 | 85.2 | 68.7 | 51.3 | 57.5 | 75.3 | +1.2 |
| | AWS [Ours] | 89.4 | 81.9 | 80.1 | 71.3 | 76.5 | 70.1 | 70.5 | 61.2 | 60.7 | 63.9 | 43.8 | 62.7 | 61.4 | 48.5 | 50.2 | 66.1 | +10.4 |
| | AWS-FS [Ours] | 90.1 | 82.9 | 81.1 | 73.1 | 77.2 | 71.8 | 71.2 | 62.7 | 62.6 | 64.9 | 46.0 | 63.6 | 62.2 | 51.0 | 51.7 | 67.4 | +9.1 |
| iBOT (Acc. 65.9%) | No Adapt [Baseline] | 86.1 | 84.2 | 86.9 | 69.3 | 87.6 | 74.6 | 73.3 | 62.3 | 62.5 | 60.3 | 36.1 | 78.5 | 62.2 | 48.9 | 47.2 | 68.0 | 0.0 |
| | Tent [ICLR'21] | 86.1 | 84.0 | 87.2 | 68.8 | 88.4 | 71.3 | 71.2 | 60.5 | 61.3 | 60.3 | 36.3 | 79.4 | 63.2 | 47.1 | 48.0 | 67.5 | +0.5 |
| | CoTTA [CVPR'22] | 86.1 | 84.3 | 87.0 | 69.3 | 87.6 | 77.3 | 73.3 | 61.8 | 61.9 | 60.3 | 36.1 | 78.0 | 61.9 | 48.4 | 46.7 | 68.0 | +0.0 |
| | SAR [ICLR'23] | 85.7 | 83.2 | 85.1 | 68.8 | 87.9 | 70.9 | 71.3 | 60.0 | 61.1 | 60.3 | 36.2 | 78.3 | 62.7 | 47.1 | 47.7 | 67.1 | +0.9 |
| | PETAL [CVPR'23] | 86.1 | 84.3 | 87.0 | 69.3 | 87.6 | 77.3 | 73.3 | 61.6 | 61.8 | 59.9 | 36.0 | 77.9 | 61.9 | 48.3 | 46.7 | 67.9 | +0.1 |
| | COME [ICLR'25] | 86.2 | 84.2 | 87.0 | 69.2 | 87.6 | 74.5 | 73.3 | 62.4 | 62.5 | 60.3 | 36.2 | 78.4 | 66.2 | 48.9 | 47.1 | 68.0 | +0.0 |
| | AWS [Ours] | 56.4 | 51.5 | 53.4 | 53.3 | 55.0 | 52.5 | 48.5 | 46.3 | 48.1 | 46.6 | 34.8 | 47.4 | 44.6 | 40.5 | 42.8 | 48.1 | +19.9 |
| | AWS-FS [Ours] | 58.2 | 53.3 | 55.2 | 55.6 | 56.0 | 54.3 | 50.8 | 48.7 | 49.7 | 48.4 | 36.4 | 49.8 | 45.8 | 42.3 | 44.3 | 49.9 | +18.1 |

Table 13: Full-results for CIFAR10-to-CIFAR10C under CTTA scenario. Mean (%) denotes the average error rate across 15 target domains.

| Pretrained Model (Src Acc. %) | Method | Gaussian | shot | impulse | defocus | glass | motion | zoom | snow | frost | fog | brightness | contrast | elastic_trans | pixelate | jpeg | Mean↓ | Gain↑ |
|---|---|---|---|---|---|---|---|---|---|---|---|---|---|---|---|---|---|---|
| Source Pretrained (Acc. 97.1%) | No Adapt | 60.1 | 53.2 | 38.3 | 19.9 | 35.5 | 22.6 | 18.6 | 12.1 | 12.7 | 22.8 | 5.3 | 49.7 | 23.6 | 24.7 | 23.1 | 28.2 | 0.0 |
| | Tent [ICLR'21] | 57.7 | 56.3 | 29.4 | 16.2 | 35.3 | 16.2 | 12.4 | 11.0 | 11.6 | 14.9 | 4.7 | 22.5 | 15.9 | 29.1 | 19.5 | 23.5 | +4.7 |
| | CoTTA [CVPR'22] | 58.7 | 51.3 | 33.0 | 20.1 | 34.8 | 20.0 | 15.2 | 11.1 | 11.3 | 18.5 | 4.0 | 34.7 | 18.8 | 19.0 | 17.9 | 24.6 | +3.6 |
| | SAR [ICLR'23] | 54.1 | 47.6 | 38.0 | 19.9 | 34.8 | 22.6 | 18.6 | 12.1 | 12.7 | 22.8 | 5.3 | 39.9 | 23.6 | 24.7 | 23.1 | 26.6 | +1.6 |
| | PETAL [CVPR'23] | 59.9 | 52.3 | 36.1 | 20.1 | 34.7 | 19.4 | 14.8 | 11.5 | 11.2 | 17.8 | 4.4 | 29.6 | 17.6 | 19.2 | 17.3 | 24.4 | +3.8 |
| | ViDA [ICLR'24] | 52.9 | 47.9 | 19.4 | 11.4 | 31.3 | 13.3 | 7.6 | 7.6 | 9.9 | 12.5 | 3.8 | 26.3 | 14.4 | 33.9 | 18.2 | 20.7 | +7.5 |
| | Continual-MAE [CVPR'24] | 30.6 | 18.9 | 11.5 | 10.4 | 22.5 | 13.9 | 9.8 | 6.6 | 6.5 | 8.8 | 4.0 | 8.5 | 12.7 | 9.2 | 14.4 | 12.6 | +15.6 |
| | COME [ICLR'25] | 54.3 | 47.1 | 36.6 | 19.9 | 34.9 | 22.6 | 18.6 | 12.1 | 12.7 | 22.8 | 5.3 | 40.7 | 23.4 | 24.7 | 23.1 | 26.6 | +1.6 |
| | AWS [Ours] | 18.7 | 13.1 | 11.0 | 10.1 | 18.8 | 9.7 | 7.2 | 7.0 | 6.2 | 9.3 | 3.8 | 10.2 | 12.4 | 9.5 | 15.4 | 10.8 | +17.4 |
| DINO (Acc. 83.1%) | No Adapt | 74.8 | 74.5 | 67.2 | 37.0 | 53.7 | 34.7 | 28.8 | 27.7 | 32.5 | 48.6 | 15.3 | 44.2 | 37.8 | 48.9 | 39.8 | 44.3 | 0.0 |
| | Tent [ICLR'21] | 76.2 | 77.7 | 68.0 | 36.3 | 54.2 | 34.0 | 27.2 | 26.9 | 30.0 | 45.7 | 13.9 | 38.7 | 35.3 | 51.1 | 36.8 | 43.5 | +0.8 |
| | CoTTA [CVPR'22] | 74.8 | 74.5 | 67.1 | 37.1 | 53.7 | 34.7 | 28.9 | 27.6 | 32.4 | 48.5 | 15.3 | 44.1 | 37.7 | 48.8 | 39.7 | 44.3 | +0.0 |
| | SAR [ICLR'23] | 76.7 | 78.1 | 67.4 | 36.4 | 53.6 | 34.0 | 26.9 | 26.2 | 29.0 | 44.7 | 13.7 | 38.1 | 34.9 | 51.8 | 36.2 | 43.2 | +1.1 |
| | PETAL [CVPR'23] | 74.6 | 72.1 | 64.2 | 34.4 | 49.1 | 33.2 | 24.5 | 24.1 | 24.0 | 27.7 | 11.2 | 29.3 | 19.4 | 36.6 | 21.8 | 36.4 | +7.9 |
| | COME [ICLR'25] | 76.4 | 76.7 | 66.7 | 38.2 | 51.7 | 34.2 | 26.5 | 25.6 | 28.0 | 43.7 | 13.3 | 35.9 | 33.9 | 53.3 | 35.2 | 42.6 | +1.7 |
| | AWS [Ours] | 51.3 | 33.9 | 35.8 | 23.3 | 32.5 | 26.0 | 18.3 | 20.4 | 19.6 | 26.8 | 12.6 | 19.1 | 25.1 | 27.4 | 29.5 | 26.8 | +17.5 |
| | AWS-FS [Ours] | 53.2 | 35.6 | 35.8 | 25.3 | 33.9 | 27.8 | 20.0 | 21.7 | 21.0 | 29.7 | 13.5 | 20.5 | 26.2 | 28.0 | 30.3 | 28.2 | +16.1 |
| MoCo (Acc. 83.6%) | No Adapt | 66.7 | 66.2 | 64.7 | 36.3 | 50.8 | 39.9 | 31.5 | 25.8 | 32.7 | 55.9 | 14.0 | 29.9 | 42.3 | 45.4 | 31.0 | 42.2 | 0.0 |
| | Tent [ICLR'21] | 67.0 | 67.3 | 65.0 | 36.4 | 51.4 | 40.0 | 31.5 | 26.5 | 34.6 | 56.3 | 14.2 | 30.3 | 42.8 | 44.7 | 32.4 | 42.7 | -0.5 |
| | CoTTA [CVPR'22] | 66.7 | 66.2 | 64.7 | 36.3 | 50.8 | 39.9 | 31.5 | 25.8 | 32.7 | 55.9 | 14.0 | 29.9 | 42.3 | 45.4 | 31.0 | 42.2 | +0.0 |
| | SAR [ICLR'23] | 66.7 | 66.2 | 64.7 | 36.3 | 50.8 | 39.9 | 31.5 | 25.8 | 32.9 | 55.6 | 13.8 | 29.8 | 42.0 | 45.2 | 31.2 | 42.2 | +0.0 |
| | PETAL [CVPR'23] | 66.7 | 66.4 | 64.8 | 36.3 | 51.1 | 39.5 | 30.4 | 26.1 | 33.8 | 54.9 | 14.0 | 29.2 | 41.9 | 44.7 | 33.0 | 42.2 | +0.0 |
| | COME [ICLR'25] | 66.7 | 66.2 | 64.7 | 36.3 | 50.8 | 39.9 | 31.5 | 25.8 | 33.0 | 55.9 | 13.8 | 30.6 | 42.2 | 44.9 | 31.4 | 42.2 | +0.0 |
| | AWS [Ours] | 66.0 | 64.5 | 62.4 | 34.1 | 49.9 | 37.6 | 27.7 | 27.4 | 32.5 | 52.2 | 14.7 | 26.0 | 38.1 | 43.7 | 33.6 | 40.7 | +1.5 |
| | AWS-FS [Ours] | 70.1 | 68.6 | 68.2 | 37.2 | 53.9 | 39.2 | 30.6 | 29.8 | 35.1 | 54.8 | 17.6 | 30.1 | 41.7 | 45.7 | 36.0 | 43.9 | -1.7 |
| iBOT (Acc. 83.4%) | No Adapt | 75.8 | 75.4 | 70.2 | 51.1 | 50.1 | 43.3 | 39.5 | 25.5 | 29.3 | 54.7 | 16.9 | 48.7 | 38.8 | 59.3 | 42.2 | 48.0 | 0.0 |
| | Tent [ICLR'21] | 76.0 | 76.0 | 70.9 | 51.5 | 50.5 | 41.3 | 35.2 | 23.7 | 27.3 | 49.1 | 14.3 | 40.7 | 35.7 | 57.7 | 37.7 | 45.8 | +2.2 |
| | CoTTA [CVPR'22] | 72.1 | 68.4 | 68.1 | 55.9 | 47.3 | 48.4 | 46.9 | 27.8 | 24.9 | 42.6 | 19.1 | 50.4 | 36.0 | 52.8 | 37.9 | 46.6 | +1.4 |
| | SAR [ICLR'23] | 80.2 | 81.8 | 74.2 | 41.5 | 48.4 | 27.0 | 18.1 | 19.5 | 22.3 | 26.0 | 14.2 | 28.2 | 31.6 | 57.8 | 32.7 | 40.2 | +7.8 |
| | PETAL [CVPR'23] | 76.7 | 77.0 | 71.7 | 49.8 | 52.1 | 42.0 | 35.9 | 25.2 | 29.5 | 48.3 | 14.8 | 39.4 | 35.6 | 54.1 | 37.6 | 46.0 | +2.0 |
| | COME [ICLR'25] | 76.4 | 76.7 | 71.5 | 52.1 | 50.8 | 43.0 | 34.0 | 25.6 | 27.1 | 45.0 | 13.4 | 37.1 | 32.6 | 60.9 | 35.5 | 45.0 | +3.0 |
| | AWS [Ours] | 70.1 | 57.6 | 54.4 | 26.7 | 36.4 | 23.7 | 15.1 | 17.4 | 18.5 | 25.9 | 10.4 | 16.5 | 21.5 | 29.3 | 28.6 | 30.1 | +17.9 |
| | AWS-FS [Ours] | 72.2 | 60.1 | 53.5 | 27.7 | 38.5 | 25.6 | 16.5 | 19.0 | 20.5 | 29.0 | 11.0 | 17.4 | 23.3 | 29.0 | 30.6 | 31.6 | +16.4 |

Table 14: Full-results for CIFAR100-to-CIFAR100C under CTTA scenario. Mean (%) denotes the mean error rate across 15 target domains.

| Pretrained Model (Src Acc. %) | Method | Gaussian | shot | impulse | defocus | glass | motion | zoom | snow | frost | fog | brightness | contrast | elastic_trans | pixelate | jpeg | Mean↓ | Gain↑ |
|---|---|---|---|---|---|---|---|---|---|---|---|---|---|---|---|---|---|---|
| Source Pretrained (Acc. 92.6%) | No Adapt | 55.0 | 51.5 | 26.9 | 24.0 | 60.5 | 29.0 | 21.4 | 21.1 | 25.0 | 35.2 | 11.8 | 34.8 | 43.2 | 56.0 | 35.9 | 35.4 | 0.0 |
| | Tent [ICLR'21] | 53.0 | 47.0 | 24.6 | 22.3 | 58.5 | 26.5 | 19.0 | 21.0 | 23.0 | 30.1 | 11.8 | 25.2 | 39.0 | 47.1 | 33.3 | 32.1 | +3.3 |
| | CoTTA [CVPR'22] | 55.0 | 51.3 | 25.8 | 24.1 | 59.2 | 28.9 | 21.4 | 21.0 | 24.7 | 34.9 | 11.7 | 31.7 | 40.4 | 55.7 | 35.6 | 34.8 | +0.6 |
| | SAR [ICLR'23] | 39.4 | 31.0 | 19.8 | 20.9 | 43.9 | 22.6 | 19.1 | 20.3 | 20.2 | 24.3 | 11.8 | 22.3 | 35.2 | 32.1 | 30.1 | 26.2 | +9.2 |
| | PETAL [CVPR'23] | 49.2 | 38.7 | 24.1 | 26.3 | 38.2 | 25.4 | 19.4 | 21.0 | 19.3 | 26.6 | 15.4 | 31.8 | 28.3 | 26.6 | 29.5 | 28.0 | +7.4 |
| | ViDA [ICLR'24] | 50.1 | 40.7 | 22.0 | 21.2 | 45.2 | 21.6 | 16.5 | 17.9 | 16.6 | 25.6 | **11.5** | 29.0 | 29.6 | 34.7 | 27.1 | 27.3 | +8.1 |
| | Continual-MAE [CVPR'24] | 48.6 | 30.7 | 18.5 | 21.3 | 38.4 | 22.2 | 17.5 | 19.3 | 18.0 | 24.8 | 13.1 | 27.8 | 31.4 | 35.5 | 29.5 | 26.4 | +9.0 |
| | COME [ICLR'25] | 39.5 | 30.5 | 19.7 | 20.7 | 41.8 | 22.5 | 17.2 | 20.2 | 17.3 | 23.7 | 12.8 | 22.3 | 34.7 | 32.2 | 29.6 | 25.6 | +9.8 |
| | AWS [Ours] | **29.0** | **24.0** | **17.2** | **17.8** | **30.5** | **19.3** | **15.7** | **16.7** | **15.7** | **19.2** | 11.8 | **15.9** | **25.9** | **20.6** | **27.0** | **20.4** | **+15.0** |
| DINO (Acc. 61.5%) | No Adapt | 82.1 | 80.6 | 78.4 | 57.9 | 78.2 | 55.7 | 49.5 | 52.0 | 55.4 | 69.4 | 36.3 | 66.4 | 62.6 | 72.6 | 64.6 | 64.1 | 0.0 |
| | Tent [ICLR'21] | 80.8 | 78.8 | 79.0 | 58.3 | 78.1 | 55.1 | 47.9 | 50.0 | 52.1 | 67.3 | 34.2 | 63.2 | 60.6 | 76.0 | 62.0 | 62.9 | +1.2 |
| | CoTTA [CVPR'22] | 82.1 | 80.6 | 79.0 | 57.9 | 78.3 | 55.6 | 49.5 | 51.9 | 55.2 | 69.3 | 36.4 | 66.4 | 62.6 | 72.7 | 64.5 | 64.1 | +0.0 |
| | SAR [ICLR'23] | 75.3 | 66.3 | 71.0 | 56.1 | 71.2 | 51.6 | **41.2** | **41.7** | 42.7 | 49.9 | **32.1** | 48.0 | **49.5** | 75.0 | **51.4** | 54.9 | +9.2 |
| | PETAL [CVPR'23] | 81.7 | 79.0 | 78.0 | 57.2 | 70.5 | 54.1 | 48.1 | 50.9 | 46.6 | 55.0 | 41.2 | 63.1 | 51.4 | 69.5 | 56.2 | 60.2 | +3.9 |
| | COME [ICLR'25] | 79.4 | 75.9 | 76.4 | 62.9 | 75.0 | 54.8 | 47.9 | 46.9 | 49.0 | 62.5 | 33.2 | 58.3 | 58.8 | 76.8 | 58.2 | 61.1 | +3.0 |
| | AWS [Ours] | **63.7** | **52.5** | **58.8** | **51.2** | **59.7** | **51.2** | 43.9 | 44.7 | **43.2** | 51.6 | 37.5 | **45.7** | 52.7 | **49.9** | 52.9 | **50.6** | **+13.5** |
| | AWS-FS [Ours] | 66.2 | 54.8 | 60.2 | 52.6 | 62.0 | 52.9 | 45.6 | 47.0 | 45.3 | 53.6 | 39.0 | 47.9 | 53.9 | 52.0 | 54.6 | 52.5 | +11.6 |
| MoCo (Acc. 59.5%) | No Adapt | 82.8 | 81.6 | 83.3 | 58.5 | 73.0 | 60.0 | 51.7 | 52.5 | 56.0 | 75.0 | 37.4 | 56.7 | 65.6 | 69.3 | 59.0 | 64.2 | 0.0 |
| | Tent [ICLR'21] | 82.9 | 81.8 | 83.4 | 58.5 | 73.2 | 60 | 51.8 | 53.0 | 56.9 | 75.2 | 37.5 | 56.7 | 66.1 | 69.1 | 59.5 | 64.4 | -0.2 |
| | CoTTA [CVPR'22] | 82.8 | 81.6 | 91.6 | 57.6 | 72.8 | 58.9 | 50.8 | 52 | 55.5 | 74.5 | **37.1** | 55.9 | 65.0 | 69.0 | 59.9 | 64.3 | -0.1 |
| | SAR [ICLR'23] | 82.8 | 82.6 | 83.3 | 58.5 | 73.0 | 60.0 | 51.7 | 52.5 | 56.0 | 74.8 | 37.3 | 57.5 | 65.6 | 69.1 | 59.1 | 64.2 | +0.0 |
| | PETAL [CVPR'23] | 82.9 | 81.7 | 91.1 | 58.5 | 73.2 | 59.7 | 51.6 | 52.6 | 56.2 | 74.6 | 37.2 | 56.2 | 65.5 | 68.7 | 59.5 | 64.6 | -0.4 |
| | COME [ICLR'25] | 82.8 | 81.6 | 83.3 | 58.5 | 73.0 | 60.0 | 51.7 | 52.5 | 55.9 | 74.8 | 37.3 | 57.5 | 65.6 | 69.1 | 59.0 | 64.2 | +0.0 |
| | AWS [Ours] | **82.2** | **79.6** | **80.4** | **56.8** | **71.5** | **57.9** | **49.6** | **51.2** | **52.7** | **70.1** | 38.1 | **53.9** | **62.5** | **65.9** | **58.7** | **62.1** | **+2.1** |
| | AWS-FS [Ours] | 83.6 | 81.6 | 82.3 | 59.5 | 72.8 | 60.3 | 52.2 | 54.6 | 56.2 | 71.5 | 41.1 | 55.1 | 65.3 | 67.7 | 61.1 | 64.3 | -0.1 |
| iBOT (Acc. 61.0%) | No Adapt | 81.3 | 80.3 | 81.4 | 69.2 | 70.7 | 62.1 | 57.1 | 47.4 | 48.9 | 70.5 | 37.3 | 71.5 | 61.4 | 79.6 | 66.2 | 65.6 | 0.0 |
| | Tent [ICLR'21] | 78.8 | 74.3 | 76.8 | 57.7 | 64.4 | 45.5 | 38.2 | 38.0 | 38.5 | 45.9 | 29.7 | 42.9 | 49.5 | 71.3 | 47.6 | 53.3 | +12.3 |
| | CoTTA [CVPR'22] | 78.9 | 74.4 | 78.0 | 69.3 | 67.3 | 62.8 | 59.5 | 47.0 | 45.4 | 66.2 | 42.0 | 76.9 | 58.3 | 84.3 | 67.3 | 65.2 | +0.4 |
| | SAR [ICLR'23] | **74.6** | 65.2 | 73.2 | 55.7 | 62.1 | **44.8** | 38.3 | **36.9** | **36.8** | 44.9 | **29.5** | 42.5 | 48.1 | 69.4 | **45.6** | 51.2 | +14.4 |
| | PETAL [CVPR'23] | 76.5 | 67.4 | 71.6 | 57.3 | **59.6** | 52.2 | 45.1 | 47.4 | 44.7 | 54.1 | 39.5 | 71.6 | 48.5 | 55.0 | 54.4 | 56.3 | +9.3 |
| | COME [ICLR'25] | 79.7 | 76.2 | 78.8 | 66.4 | 67.7 | 56.8 | 52.1 | 41.6 | 41.3 | 56.4 | 32.5 | 62.0 | 54.7 | 80.5 | 61.3 | 60.5 | +5.1 |
| | AWS [Ours] | 75.6 | **64.9** | **67.0** | **47.6** | 60.5 | 45.9 | **37.8** | 39.9 | 39.2 | 47.7 | 31.5 | 44.5 | 48.2 | **48.5** | 51.7 | **50.2** | **+15.4** |
| | AWS-FS [Ours] | 76.3 | 67.4 | 69.3 | 49.6 | 62.5 | 47.2 | 39.4 | 42.3 | 41.2 | 49.6 | 33.7 | 45.6 | 50.1 | 50.3 | 53.4 | 51.9 | +13.7 |

