# OpenReview forum: "When Test-Time Adaptation Meets Self-Supervised Model"
_ICLR.cc/2026/Conference — ICLR 2026 Conference Withdrawn Submission_

### Official Review · Reviewer_khWj · 2025-10-19

**Soundness:** 2
**Presentation:** 3
**Contribution:** 3
**Rating:** 4
**Confidence:** 3

**Summary:**

Online test-time adaptation methods typically require a separate source-pretrained model for each target domain, which poses challenges in terms of practical efficiency and generalization. To address this limitation, the authors leverage the zero-shot capability of self-supervised models and design a distance-based classifier that relies on class prototypes obtained solely through forward passes. To enable effective test-time adaptation, they further propose a collaborative learning framework that integrates contrastive learning and knowledge distillation for progressive representation refinement. Extensive experiments demonstrate the effectiveness of the proposed approach.

**Strengths:**

1. The paper proposes leveraging self-supervised models to address the limitation that existing online test-time adaptation methods require a separate source-pretrained model for each target domain. This idea is novel and has practical significance for real-world deployment.
2. The paper is clearly written, and the figures and tables effectively convey the authors’ ideas.
3. The experimental results are comprehensive and convincingly validate the proposed approach.

**Weaknesses:**

1. It is unclear why the authors choose to freeze the classifier of the SSL model rather than that of the target model. Clarifying which modules each loss term updates, along with the underlying reasoning, would improve the clarity of the method description.
2. I suggest adding pseudocode to make the overall procedure easier for readers to follow.
3. The implementation details of the baselines are not well explained. Specifically, do the baseline methods update the self-supervised model or the classifier? The authors are encouraged to clarify this point.
4. Since test-time adaptation emphasizes adaptation efficiency rather than training cost, it would be helpful if the authors could compare the time and memory overhead of AWS with that of the baselines during the adaptation phase, to better evaluate its deployability in real-world scenarios.

**Questions:**

See weaknesses.

---

### Official Review · Reviewer_ZoAT · 2025-10-29

**Soundness:** 2
**Presentation:** 3
**Contribution:** 2
**Rating:** 4
**Confidence:** 4

**Summary:**

The paper proposes **a new setting** named Self-Supervised Test-Time Adaptation (Self-Supervised TTA). Different from previous TTA settings that start from a source-pretrained model, the new setting aims to perform TTA starting from a self-supervised learning (SSL) model.

The authors introduce **a method** named AWS (Adapt Without Source pretraining) for this setting. The method includes a framework that integrates contrastive learning, knowledge distillation, and mutual learning techniques, designed for a “no-head” SSL model architecture. Experiments on CIFAR-10-C, CIFAR-100-C, and ImageNet-C show that AWS clearly outperforms selected baselines under the proposed setting.

**Strengths:**

1. The paper introduces a new setting. It explores TTA toward target domains starting from an SSL model rather than a source-pretrained model.
2. Empirical results demonstrate that the proposed method, AWS, handles the proposed setting better than existing TTA baselines.
3. The paper is clearly written and well-structured.

**Weaknesses:**

1. The paper claims that previous TTA methods fail when applied to self-supervised models, as shown in Figure 2(a)(b). However, the observed degradation might stem from the inherently lower base accuracy of SSL models on target domains rather than an intrinsic incompatibility of TTA methods. The evidence therefore does not convincingly isolate why these methods fail. A deeper analysis of the causal factors would strengthen the motivation.
2. The AWS framework combines contrastive learning, knowledge distillation, and mutual learning — all of which have been used in prior TTA works. It is somewhat unclear why AWS succeeds in this setting with these techniques, whereas previous TTA methods using similar components fail.
3. Related to point 2, my understanding is that AWS mainly reformulates these techniques to fit the SSL model architecture. Specifically, since an SSL model does not have a classifier head like source-pretrained models, AWS adapts these techniques accordingly. Therefore, can the performance gains be primarily attributed to the algorithmic adjustment to the model architecture rather than to a new learning principle?
4. Recent TTA works such as LAME [a], NOTE [b], RoTTA [c], SAR, and DA-TTA [d] emphasize adaptation under non-i.i.d. data streams — arguably a more realistic setting. Since the proposed Self-Supervised TTA also aims to move toward practical TTA scenarios, a discussion of non-i.i.d. settings (e.g., whether AWS can handle temporal correlations) would make the contribution more comprehensive.

[a] Parameter-free online test-time adaptation. CVPR 2022.
[b] Note: Robust continual test-time adaptation against temporal correlation. NeurIPS 2022.
[c] Robust test-time adaptation in dynamic scenarios. CVPR 2023.
[d] Distribution alignment for fully test-time adaptation with dynamic online data streams. ECCV2024.

**Questions:**

Related to the **Weaknesses**:

- Could the authors provide a more direct analysis or ablation to disentangle whether the failure of conventional TTA methods on SSL models (Figure 2) arises from the inherently low base accuracy of SSL models or from a fundamental incompatibility with SSL representations?
- How do the proposed contrastive, distillation, and mutual information losses differ mathematically or functionally from those in existing TTA methods?
- Is the main improvement primarily driven by adapting these objectives to the SSL model architecture (which lacks a classifier head), or by new algorithmic insights beyond architectural compatibility?
- Could AWS handle a practical non-i.i.d. test stream, as in LAME, RoTTA, or DA-TTA?


**Additional (non-evaluative) question:**
On a more general and philosophical note — this setting aims to adapt a self-supervised model to any test domain in an online, unsupervised manner. Considering the model’s full lifecycle (trained in SSL and adapted without any labeled supervision), do the authors think this goal might be too strict or idealized in practice? I am simply curious about how the authors view the feasibility and long-term vision of such a “fully unsupervised lifelong” adaptation paradigm.

---

### Official Review · Reviewer_58Wy · 2025-10-31

**Soundness:** 3
**Presentation:** 2
**Contribution:** 3
**Rating:** 6
**Confidence:** 4

**Summary:**

This paper investigates the integration of Self-Supervised Learning (SSL) with Test-Time Adaptation (TTA) methods, introducing a Self-Supervised TTA framework (AWS) that combines contrastive learning, knowledge distillation, and mutual learning. The experiments demonstrate improvements in domain adaptation performance over existing methods, particularly in the context of SSL models like DINO, MoCo, and iBOT. The AWS method outperforms traditional pretraining-based models on various benchmarks, including CIFAR10 and ImageNetC.

**Strengths:**

1. Innovative Problem Focus: The paper addresses the significant challenge of test-time adaptation for self-supervised models, a promising area of research.
2. Clear Performance Improvement: The proposed method demonstrates substantial performance gains, especially in adapting self-supervised models without relying on source pretraining.
3. Visual Clarity: Figures are well-constructed, providing clear visual support for the concepts and experimental results discussed.

**Weaknesses:**

1. **Prototype Construction Ambiguity**: In Table 1, while it is claimed that no training labels are required, the method for constructing prototypes is unclear. Specifically, how are class prototypes created without relying on source or target labels? This could be misleading, as it seems that some form of supervision is still required (likely source labels). Clarifying this aspect would reduce potential confusion.

2. **Complexity and Loss Balancing**: The proposed mechanism is quite complex, involving multiple loss components. Balancing the weights of these losses (λ parameters) may require careful tuning, which introduces an inherent limitation. Simplifying the approach or providing more guidance on selecting appropriate weight values for the different losses could make the method more practical for implementation.

3. **Similarity with Entropy Minimization**: The mechanism of prototype-based classification, which relies on cosine similarity followed by softmax, inherently reduces entropy by pulling the predicted class closer while pushing others away. This seems to overlap with entropy minimization. If AWS is essentially a variant of entropy minimization with regularization, it would be helpful to better explain why it significantly outperforms Tent, particularly if both use similar principles. Additionally, clarity on whether learning rates were systematically explored using [1e-3, 1e-4, 1e-5, 1e-6] × batch size / 64 could provide insight into whether the experimental settings fully support the claims.

**Questions:**

Clarification on Line 73: The paper mentions that "SSL models, where low entropy does not ensure accurate predictions." Could you elaborate on why low entropy cannot ensure accurate predictions in self-supervised models? What is the key difference between SSL and other models in this regard, and how does it affect performance in TTA? A deeper explanation here would clarify an important aspect of your approach.

---

### Official Review · Reviewer_fyxF · 2025-11-01

**Soundness:** 3
**Presentation:** 2
**Contribution:** 2
**Rating:** 4
**Confidence:** 5

**Summary:**

This paper explores integrating self-supervised learning (SSL) models into test-time adaptation (TTA) to address domain shifts without relying on source-domain pretraining. Traditional TTA methods depend on supervised source models, but the authors propose a Self-Supervised TTA protocol using SSL models (e.g., DINO, MoCo, iBOT) and a prototype classifier built via forward passes only. They introduce the AWS framework, combining contrastive learning for representation refinement, knowledge distillation to preserve generalization, and mutual learning for collaborative adaptation.

**Strengths:**

1. Novel integration of SSL into TTA, reducing computational costs by eliminating source-specific pretraining while maintaining competitive performance.

2. Efficient prototype classifier avoids backpropagation, enabling quick adaptation (e.g., few-shot with 30 samples/class yields near full-shot results).

3. Comprehensive evaluation across multiple SSL backbones, benchmarks, and settings (e.g., natural shifts like ImageNet-R, semantic segmentation on Cityscapes-ACDC), demonstrating robustness and gains (e.g., +16.4% on ImageNetC for source-pretrained).

**Weaknesses:**

1. Table 1 lacks clarity: Terms like "Stationary" vs. "Continually changing" are not defined; the distinction from TTT (Sun et al., 2020), which also uses self-supervision, is unclear—key differences in protocol and objectives should be explicitly stated.
2. Eq(5) design is under-justified: The first term uses mutual information while the second uses cross-entropy; no rationale is provided for this asymmetric loss combination, weakening theoretical grounding.
3. Method appears incremental: Core contribution is largely adapting existing SSL losses (contrastive, distillation) to TTA without deeply engaging TTA-specific challenges (e.g., online stability, error accumulation); lacks novel insights into continual domain shifts.
4. Baseline setup ambiguous in main body: It is unclear whether TTA baselines (e.g., Tent, CoTTA) are applied to supervised or SSL-pretrained models when evaluating SSL backbones—critical for interpreting gains. SSL pretraining method (e.g., DINO vs. MoCo) is also not specified upfront in the main text.

**Questions:**

Please refer to Weaknesses.

---

### Note · Authors · 2025-11-13

I have read and agree with the venue's withdrawal policy on behalf of myself and my co-authors.